# Photocatalytic *Z/E* isomerization unlocking the stereodivergent construction of axially chiral alkene frameworks

Jie Wang[1,2], Jun Gu[1,2], Jia-Yu Zou[1], Meng-Jie Zhang[1], Rui Shen[1], Zhiwen Ye[1], Ping-Xun Xu[1] & Ying He ◎[1] ✉

The past century has witnessed a large number of reports on the *Z/E* isomerization of alkenes. However, the vast majority of them are still limited to the isomerization of di- and tri-substituted alkenes. The stereospecific *Z/E* isomerization of tetrasubstituted alkenes remains to be an underdeveloped area, thus lacking in a stereodivergent synthesis of axially chiral alkenes. Herein we report the atroposelective synthesis of tetrasubstituted alkene analogues by asymmetric allylic substitution-isomerization, followed by their *Z/E* isomerization via triplet energy transfer photocatalysis. In this regard, the stereodivergent synthesis of axially chiral *N*-vinylquinolinones is achieved efficiently. Mechanistic studies indicate that the benzylic radical generation and distribution are two key factors for preserving the enantioselectivities of axially chiral compounds.

Alkenes are fundamental building blocks since their various applications in medicinal and material chemistry, and as chemical platforms in the field of organic synthesis[1]. In this context, stereoselective synthesis of alkenes in both *Z*- and *E*-configurations presents a big challenge in organic synthesis[2–5]. An elegant method to achieve this goal is alkene *Z/E* isomerization which relies upon the unique properties of the C=C bond. Indeed, the *Z/E* isomerization of alkenes can be traced back to 1923 by Herzig and Faltis[6]. After that, synthetic strategies of this transformation is improving by leaps and bounds, including photocatalysis, acid (or base) catalysis, and metal catalysis[7–13]. Despite these advances, the *Z/E* isomerization are mainly limited to the di- and tri-substituted alkenes (Fig. 1A). In contrast, the isomerization of tetra-substituted alkenes is more challenging and less reported (Fig. 1B)[14–16]. The pioneering work in this area was the reports on molecular motors by Nobel laureate B. Feringa[17–21]. In this regard, the photoinduced *cis-trans* isomerizations of helical alkenes with a 180º rotation around the C=C bond were established. The molecular motor would undergo 360º rotation by combining the photochemical and thermal isomerization. Herein the point chiral elements that existed in the molecular motor are one of the essential factors for this tetrasubstituted alkene isomerization. In 2020, another seminal work on *Z/E* isomerization of

boron-containing tetrasubstituted alkenes by photocatalysis was reported. A subtle 90º C(sp²)-B bond rotation intermediate was proposed to achieve the directional *E* to *Z* isomerization[14]. Nonetheless, the stereospecific *Z/E* isomerization of acyclic axially chiral alkene analogs remains elusive (Fig. 1B).

Axially chiral alkenes was first mentioned and proposed by Kawabata in 1991 during the research on the memory of chirality[22]. It is only in recent years that studies for their catalytic synthesis were initiated[23–25]. Accordingly, several typical synthetic methods have been developed to date[26–33]. Among these, asymmetric allylic substitution-isomerization (*AASI*) which was developed by our group, has emerged as an appealing strategy for the synthesis of axially chiral alkene analogs[34–38]. Unlike other strategies relying on very complex specific substrates, *AASI* can provide a versatile and modular way to accomplish the construction by using easily accessible allylic carbonate electrophiles and sterically bulky nucleophiles. Inspired by *AASI* and alkene *Z/E* isomerization, we envision that the stereodivergent synthesis of axially chiral tetrasubstituted alkenes could be achieved by combining *AASI* with geometrical *Z/E* isomerization. Ideally, such axially chiral *Z* (or *E*) alkene could be isomerized to its corresponding *E* (or *Z*) configuration while maintaining the enantioselectivity.

---

[1]School of Chemistry and Chemical Engineering, Nanjing University of Science and Technology, Nanjing 210094, China. [2]These authors contributed equally: Jie Wang, Jun Gu. ✉e-mail: yhe@njust.edu.cn

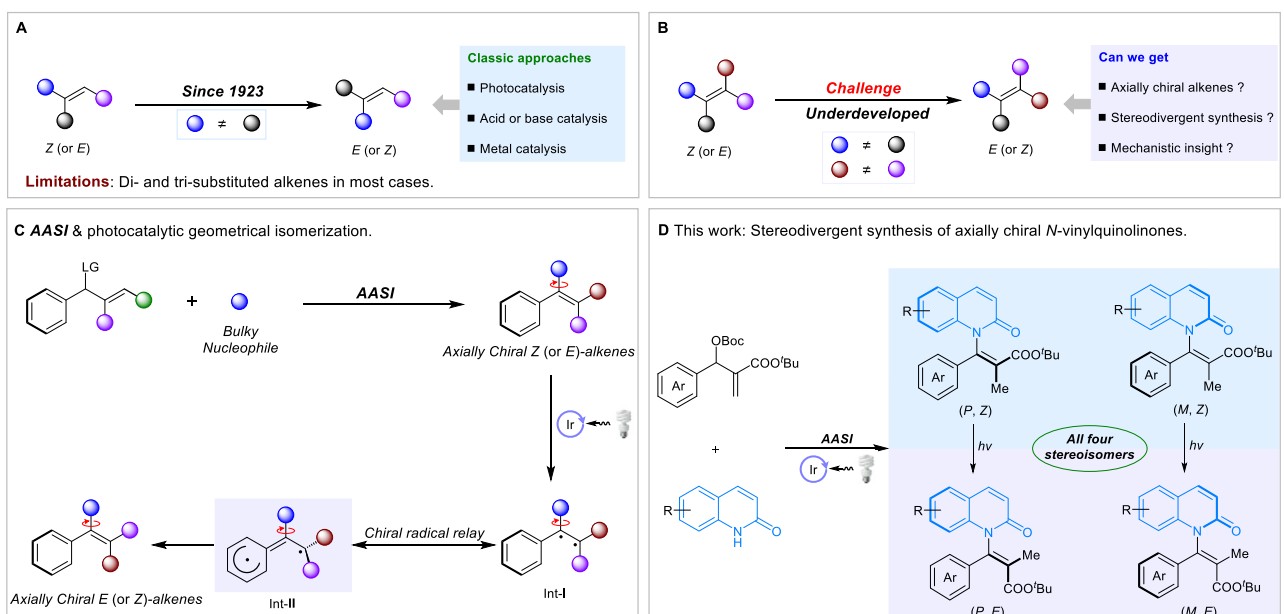

**Fig. 1 | State of the art for the stereodivergent synthesis of axially chiral alkene analogs. A** $Z/E$ isomerization of di- and tri-substituted alkenes. **B** $Z/E$ isomerization of tetrasubstituted alkenes. **C** Asymmetric allylic substitution-isomerization and photocatalytic geometrical $Z/E$ isomerization. **D** Stereodivergent synthesis of axially chiral *N*-vinylquinolinones.

To this end, we posit that the $Z$ axially chiral styrene analogs, which are synthesized by *AASI* herein, could isomerize to their corresponding $E$ products by a photocatalytic isomerization. As shown in Fig. 1C, the ground-state alkenes could be excited to a higher energy triplet state radical by energy transfer (EnT) process[39–43], thus leading to the intermediate **I**. The phenyl ring would distribute the benzyl radical with the chiral radical relay[44,45], which affords the transient axially chiral intermediate **II**. Afterward, the bond rotation occurs while preserving the axial chirality around the C-N axes. At last, the radical recombination would give the corresponding $E$ axially chiral styrene analogs. However, in our own assessment, there are still several challenges that are addressed to achieve our goal: (1) as for the *AASI* process, the nucleophiles are sterically bulky that may cause the low reactivity of reaction. In addition, the products must be obtained with high enantioselectivity, regioselectivity and geometrical ($Z/E$) selectivity simultaneously; (2) as for the photocatalytic isomerization process, the initial excitation via energy transfer creates a triplet biradical that may permit free rotation around the chiral axis, thus leading to the loss of axially chiral information.

Herein, we report the atroposelective synthesis of axially chiral *N*-vinylquinolinones by *AASI* and photocatalytic geometrical $Z/E$ isomerization that leads to the stereodivergent synthesis of tetrasubstituted alkenes (Fig. 1D).

## Results and discussion

We commenced our studies by investigating the chiral Lewis base-catalyzed asymmetric allylic substitution (*AAS*) of Morita–Baylis–Hillman (MBH) esters **1a** with 2(1*H*)-quinolinone **2a**. The reaction optimization revealed that the enantioenriched β-substituted allylic intermediate (**Int-3a**) was first generated in high yield and enantioselectivity when using (DHQD)$_2$PYR as the catalyst. Afterward, the use of inorganic base MeONa promoted the isomerization effectively that afforded axially chiral product **3a** in high yield and enantiospecificity (es) with $Z/E$ ratio beyond 19/1 (see Supplementary Fig. 2). It should be noted that the two-step processes could be carried in one-pot which gave the product (*P,Z*)-**3a** in 87% yield and 94% enantiomeric excess (ee) (Fig. 2). With the optimized conditions in hand, we then explored the scope of the reaction. Pleasingly, a wide range of aryl MBH esters bearing *para-* and *meta*-substituents were accommodated, affording (*P,Z*)-(**3b**-**3j**) in 74-92% yield with 88–95% ee. Furthermore, a number of fused-rings and heterocycles were well-tolerated, including a naphthalene [(*P,Z*)-**3k**], a thiophene [(*P,Z*)-(**3l**)] and a benzothiophene [(*P,Z*)-(**3m**)]. With respect to the 2(1*H*)-quinolinone scope, various substituents at different sites on quinolinone ring were then studied (Fig. 2). The reaction of 2(1*H*)-quinolinones bearing electron-donating and electron-withdrawing groups at the C4 position proceeded smoothly, affording products (*P,Z*)-(**3n**-**3r**) in 65–79% yield with 92–96% ee. Moreover, substituents attached at the C5, C6 or C7 sites were all compatible with the reaction conditions, affording axially chiral *N*-vinylquinolinones (*P,Z*)-(**3s**-**3x**) in 56-84% yields with 93–97% ee. The high compatibility of the reaction system encouraged us to investigate its practicality for late-stage functionalization. In this context, substrates derived from *L*-menthol and vitamin E underwent the *AASI* smoothly, giving the products (*P,Z*)-**3y** and (*P,Z*)-**3z** in 72% and 64% yield with >20/1 diastereomeric ratio (dr), respectively. These results offer a general and promising approach for the functionalization of late-stage molecules to axially chiral compounds. Of note, in all cases, axially chiral *N*-vinylquinolinones **3** were obtained in high $Z$-selectivities. The absolute configuration of (*P,Z*)-**3a** and (*P,Z*)-**3m** were confirmed by single-crystal X-ray analysis, and others were assigned by analogy.

Having established the strategy for the synthesis of **3** in $Z$ configuration, we investigated the feasibility of photocatalytic geometrical $Z/E$ isomerization of (*P,Z*)-**3** to (*P,E*)-**3**. If successful, it would represent a rare example of accessing axially chiral styrene analogs in both $E$ and $Z$ stereoisomers, thus providing the possibility for the stereodivergent synthesis of alkenes. As shown in Table 1, we were pleased to find that (*P,Z*)-**3a** could isomerize to (*P,E*)-**3a** in the presence of photosensitizer *fac*-Ir(ppy)$_3$ ($E_T$ = 58.1 kcal mol$^{-1}$) under the irradiation of 420 nm light-emitting diodes (LED)[46,47]. Although only moderate $E/Z$ ratio was observed, (*P,E*)-**3a** was efficiently afforded in 93% ee (99% es). The absolute configuration of (*P,E*)-**3a** were unambiguously determined by single-crystal X-ray analysis. This result revealed the high efficiency of the geometric $Z/E$ isomerization of tetrasubstituted alkene. Other photosensitizers such as Ru(bpy)$_3$Cl$_2$·6H$_2$O and 2,3,5,6-tetra-kis(carbazol-9-yl)−1,4-dicyanobenzene (4CzTPN) was proved to be

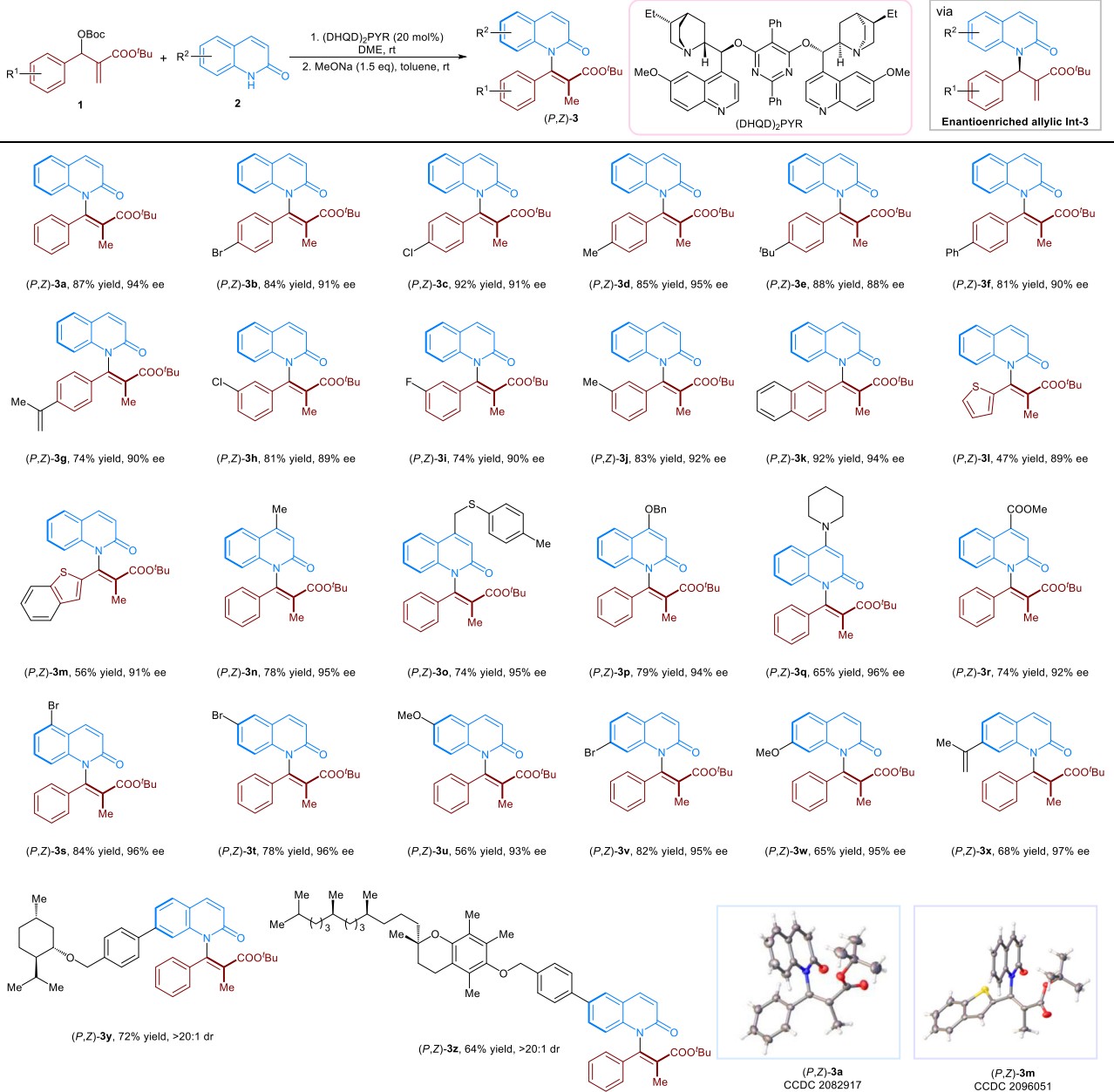

**Fig. 2 | Substrate scope of *AASI*.** Reaction conditions, **1** (0.9 mmol), **2** (0.3 mmol), (DHQD)₂PYR (20 mol%), 1,2-dimethoxyethane (DME) (1.5 mL), rt for 48 h. NaOMe (0.45 mmol), and toluene (1.0 mL) was then added after the rotary evaporation of DME. Isolated yield, the ee values were calculated by chiral HPLC traces. The dr ratios were determined by crude ¹H-NMR. The *Z/E* ratios of the products were obtained beyond 19/1.

ineffective (Table 1, entries 2 and 3). The use of {Ir(dtbpy)[dF(CF₃) ppy]₂}PF₆ gave us a moderate *E/Z* ratio with 98% es (entry 4). The examination of the solvent effect suggested that tetrahydrofuran (THF) is the best choice for enhancing the selectivity (*E/Z* = 8/1) (entries 5–7). In this case, (*P,E*)-**3a** was readily isolated in 94% ee (100% es). In addition, no *Z/E* isomerization occurred in the absence of light or photosensitizer (entry 8). To rule out the thermal background reactivity, the reaction was carried out in THF at 60 °C in the absence of the photosensitizer. As a result, no product of (*P,E*)–**3a** was observed (entry 9).

We then investigated the substrate scope of photochemical *Z/E* isomerization. As shown in Fig. 3, the reactions proceeded smoothly when using (*P,Z*)–**3** possessing phenyl moiety with *para-* and *meta-* substituents. In this regard, (*P,E*)-(**3b**–**3j**) were obtained in 64–86% isolated yields and 99-100% es with *E/Z* ratio up to 15/1. The reaction

also tolerated substrates bearing the fused and heterocyclic ring, affording (*P,E*)-(**3k**–**3m**) in good yields and enantiospecificities with moderate to good *E/Z* ratios. Different substituents attached at the C4, C5, or C6 sites of the quinolinone ring were also examined. As a result, (*P,E*)-(**3n**-**3v**) were afforded good to excellent yields and *E/Z* ratios with high es except (*P,E*)-**3r**. We speculated that the low enantioselectivity of (*P,E*)-**3r** is attributed to the undesired radical fragmentation via the quinolinone ring rather than the phenyl ring. In addition, (*P,Z*)-**3w** were accommodated to the reaction conditions which afforded (*P,E*)-**3w** in excellent yield, ee and *E/Z* ratio. However, the reaction was less reactive when using substrate bearing bulky group at C7 site of the quinolinone ring [(*P,E*)-**3x**]. Of particular note, in all cases, the (*P,Z*)-**3** in the reaction system were recovered without any loss in their enantioselectivities. In addition, the late-stage molecules were also compatible to give (*P,E*)-**3y** and (*P,E*)-**3z** in moderate yields and beyond 20/1 dr.

## Table 1 | Optimization of photochemical Z/E isomerization

(P,Z)-3a, 94% ee → fac-Ir(ppy)₃ (1 mol%), MeCN (1.0 mL), rt, 18 h, 20 W, 420 nm LED → (P,E)-3a

CCDC 2252177

| Entry | Variation from "standard conditions" | E/Z | ee (%) | es (%) |
|---|---|---|---|---|
| 1 | None | 1/1 | 93 | 99 |
| 2 | Ru(bpy)₃Cl₂ 6H₂O as the photosensitizer | - | - | - |
| 3 | 4CzTPN as the photosensitizer | - | - | - |
| 4 | {Ir(dtbpy)]dF(CF₃)ppy]₂PF₆ as the photosensitizer | 3/1 | 92 | 98 |
| 5 | Toluene as the solvent | 1/1 | 93 | 99 |
| 6 | MeOH as the solvent | 3/1 | 94 | 100 |
| 7 | THF as the solvent | 8/1 | 94 | 100 |
| 8 | No light or no photosensitizer | - | - | - |
| 9 | No photosensitizer, 60 °C in THF | - | - | - |

Reaction conditions, (P,Z)-3a (0.1 mmol), photosensitizer (1 mol%), solvent (1.0 mL), rt for 18 h. The ee values of (P,E)-3a were calculated by chiral HPLC traces. The E/Z ratio of 3a was calculated by crude $^1$H-NMR after the reactions. Es (enantiospecificity) = [ee$_{(P,E)}$-3a/ee$_{(P,Z)}$-3a]*100%.

With the synthetic advances of our strategy demonstrated, we moved forward to gain insights into photocatalytic Z/E isomerization of (P,Z)-3a. As shown in Fig. 4A, a triplet EnT photocatalytic mechanism was proposed. Photosensitization of (P,Z)-3a by the excited states Ir(ppy)₃ generates the activated diradical intermediate I. The rapid benzyl radical distribution occurs to resonate with the key intermediate II. In this regard, the high efficiency of enantiospecificity could be anticipated since II is still axially chiral intermediates around the C-N axis. The C-C bond of intermediate II would be twisted by 90°, which isomerized to the chiral intermediate III. The radical redistribution of III resonated with the intermediate IV would undergo the spin crossover to intermediate V, followed by the radical recombination to furnish the product (P,E)-3a.

We subsequently carried out the time-dependent density functional theory (TDDFT) calculations to study the reaction mechanism[9,46-50]. As shown in Fig. 4B, compared to the energy of (P,Z)-3a, (P,E)-3a is 3.4 kcal/mol lower than (P,Z)-3a. The vertical excitation energies (S₀ → T₁) for (P,Z)-3a and (P,E)-3a were calculated to be 54.5 and 55.9 kcal/mol, respectively, indicating that (P,Z)-3a was more easily activated by the triplet EnT than that of (P,E)-3a. The chiral intermediate III would be generated by EnT photocatalysis with the energy of 41.7 kcal/mol (See Supplementary Fig. 6). Due to the axially chiral character of intermediate III, in this case, (P,E)-3a was afforded with high enantiospecificity during the isomerization process.

Encouraged by the aforementioned results, we then investigated the stereodivergent synthesis of axially chiral N-vinylquinolinones 3. To our delight, besides the generation of (P,Z)-3a and (P,Z)-3s (Fig. 5, a), the one-pot synthesis of (P,E)-3a and (P,E)-3s were evaluated which shown the competitive yields and ee (Fig. 5, b). By using (DHQ)₂PYR as the catalyst under similar reaction conditions, (M,Z)-3a and (M,Z)-3s were obtained in both 78% yields with 86% and 88% ee, respectively (Fig. 5, c). Combining AASI and photocatalytic isomerization, (M,E)-3a and (M,E)-3s were readily afforded in 65% yield with 84% ee and 61% yield with 88% ee, respectively (Fig. 5, d).

Considering the potential application of atropisomers in pharmaceutical chemistry[51,52], the stereodivergent transformations of axially chiral N-vinylquinolinones 3a and 3v were performed (Fig. 6A). First, reduction of 3a gave all four stereoisomers (P,Z)-4, (M,Z)-4, (P,E)-4 and (M,E)-4 in moderate to good yields while maintaining the corresponding enantioselectivities. In addition, the double addition of 3a with methylmetallic reagents afforded compounds 5 in all four configurations with high enantiospecificities, from which (P,Z)-5 was unambiguously confirmed by its single-crystal X-ray analysis. On the other hand, the Sonogashira coupling of axially chiral N-vinylquinolinones 3v with phenylacetylene generated the (P,Z)-6 and (P,E)-6 in excellent yields and enantioselectivities (Fig. 6B). In addition, compounds 3v could be well-tolerated to the sequential Sonogashira coupling and base-promoted deprotection which gave the (P,Z)-7 and (P,E)-7 in 86% and 93% ee, respectively.

At last, we performed the racemization experiments of axially chiral N-vinylquinolinones 3. As shown in Fig. 7, comparison of rotational barriers around the C-N axis (Z-3a vs E-3a) indicated that the E-3a (27.0 kcal/mol) has a higher configurational stability than Z-3a (25.7 kcal/mol). These results indicate that the steric hindrance of the tetrahedral methyl group may be bigger than that of the ester group. Although the configurational stability of 3 is not very high, the easy transformation of the ester group in 3 would give axially chiral products in high configurational stabilities. For example, as shown in Fig. 6A, the rotational barriers of Z-4 and Z-5 in toluene are 31.3 and 33.5 kcal/mol, respectively.

In conclusion, we have disclosed the highly efficient photocatalytic Z/E geometrical isomerization of axially chiral alkene analogs. Although alkene geometrical Z/E isomerization has been realized for one century, the stereodivergent synthesis through AASI and photocatalytic Z/E isomerization for accessing tetrasubstituted alkenes

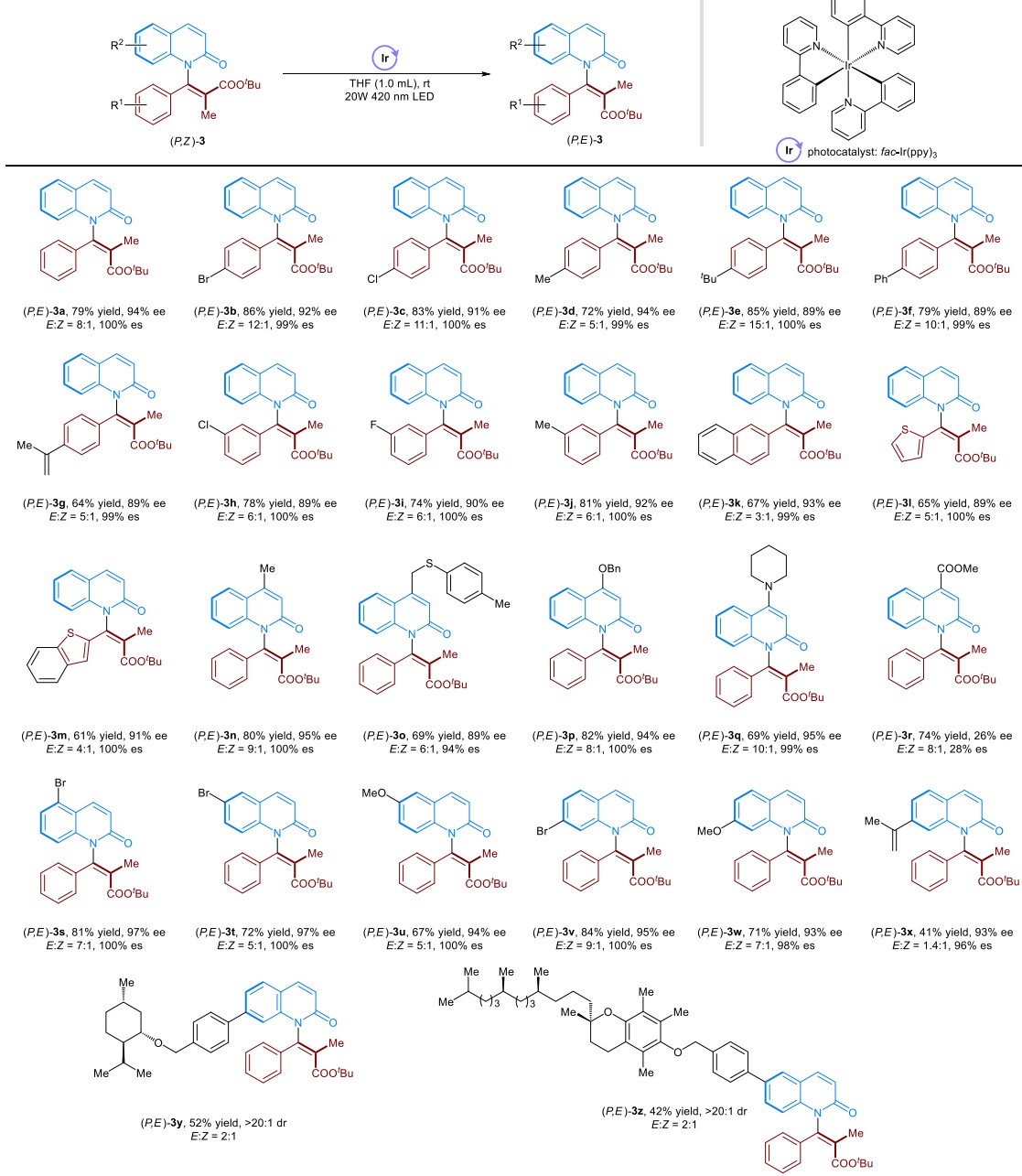

**Fig. 3 | Substrate scope of photochemical Z/E isomerization.** Reaction conditions, (*P,Z*)-**3** (0.1 mmol), *fac*-Ir(ppy)₃ (1 mol%), THF (1.0 mL), rt for 18 h, isolated yield of (*P,E*)-**3**. The ee values of (*P,E*)-**3** were calculated by chiral HPLC traces. The *E/Z* and dr ratios of **3** were calculated by crude ¹H-NMR after the reactions. Es (enantiospecificity) = [ee$_{(P,E)\text{-}3}$/ee$_{(P,Z)\text{-}3}$]*100%.

herein uncovers the cognition that the axial chirality could be transferred between *Z*- and *E*-alkenes while maintaining the enantioselectivities. We anticipate that this strategy may be adapted to a wide variety of axially chiral molecules in stereodivergent fashion.

## Methods
### General procedure for the synthesis of axially chiral products (*P,Z*)-3
The catalyst (DHQD)₂PYR (0.06 mmol, 52.8 mg), **1** (0.9 mmol), **2** (0.3 mmol) were added to a 2-dram scintillation vial equipped with a magnetic stirring bar. The vial was then charged with DME (1.5 mL) and stirred at room temperature for several days. The reaction was monitored by thin-layer chromatography (TLC) analyses. When the reaction was completed, the solvent in the vial was removed by distillation under reduced pressure. Then MeONa (0.45 mmol, 1.5 eq) was added into the vial. The vial was then charged with toluene (1.0 mL) and stirred at room temperature for about 1 h until the full consumption of the intermediate by TLC monitoring. Then the residue was purified directly by column chromatography over silica gel (petroleum ether: ethyl acetate = 10:1 to 5:1) to afford the desired product (*P,Z*)-**3**.

### General procedure for the photocatalytic synthesis of axially chiral products (*P,E*)-3
Compound (*P,Z*)-**3** (0.1 mmol, 1.0 eq) and Ir(ppy)₃ (1 mol%) were weighed out into a 2-dram scintillation vial. The vial was charged with THF (1.0 mL) and the reaction was stirred at room temperature under visible light irradiation (420 nm) for 18 h. Then the mixture was concentrated under reduced pressure and purified directly by column

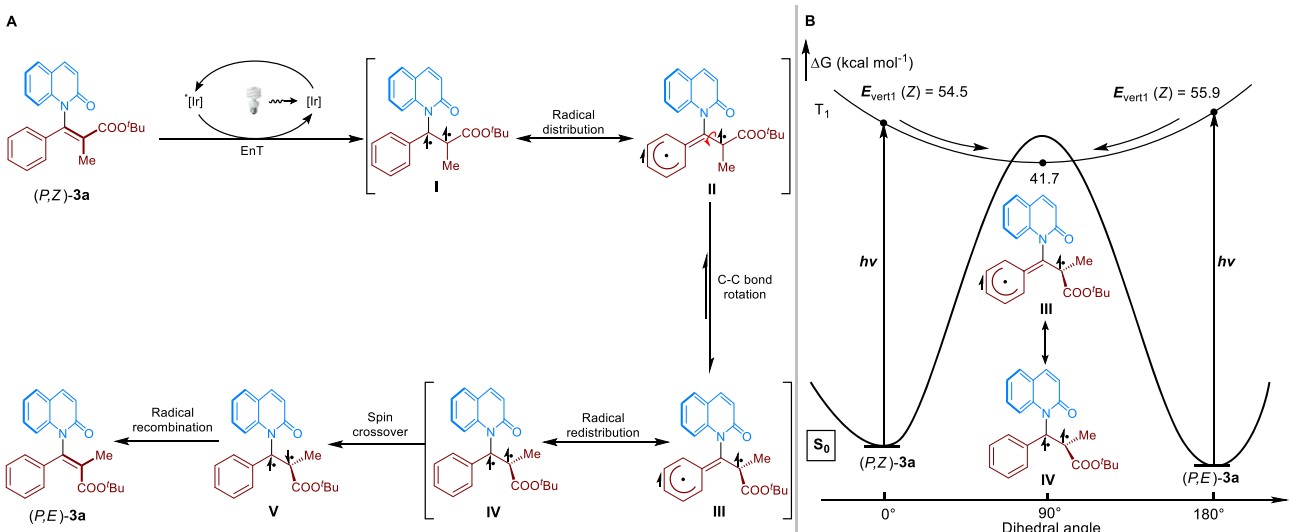

**Fig. 4 | Mechanistic studies. A** Proposed mechanism of *Z/E* isomerization of **3a**. **B** TDDFT studies of *Z/E* isomerization of **3a**. The Gibbs free energy (in kcal/mol) profiles for the photocatalytic *Z/E* isomerization and the optimized structures of critical intermediates. All energies were calculated at the M06-2X/def2-TZVPP(SMD)// M06-2X/def2-SVP(SMD).

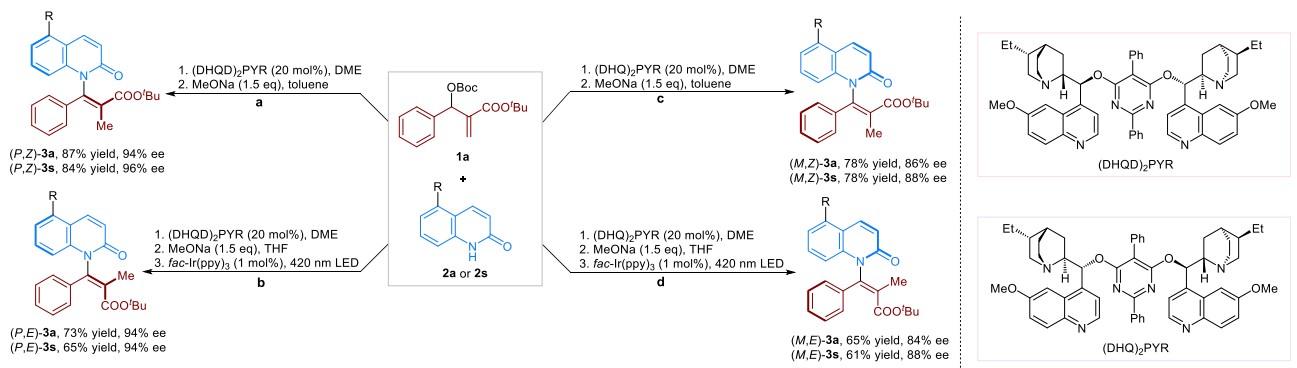

**Fig. 5 | Stereodivergent synthesis.** Stereodivergent synthesis of axially chiral *N*-vinylquinolinones **3a** and **3s**.

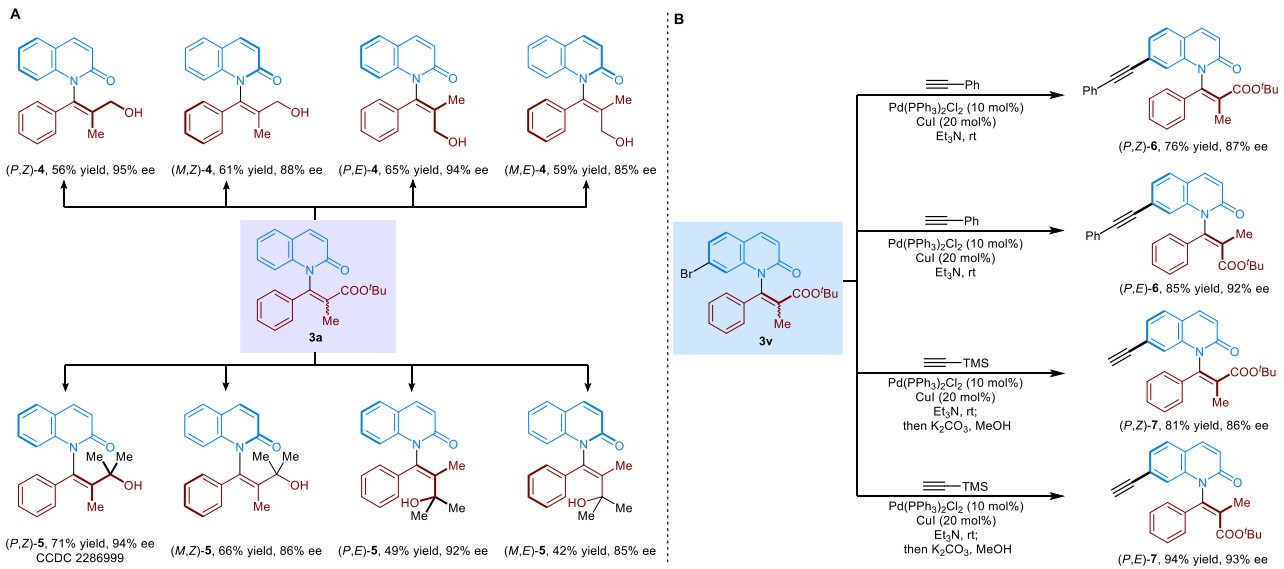

**Fig. 6 | Product transformations. A** Stereodivergent transformations of axially chiral *N*-vinylquinolinones **3a**. **B** Stereodivergent transformations of axially chiral *N*-vinylquinolinones **3v**.

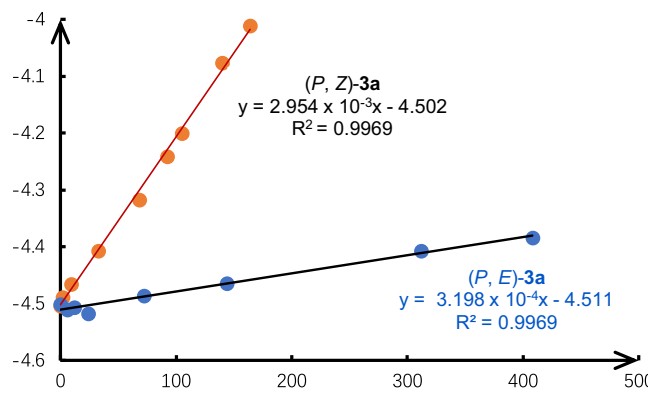

**Fig. 7 | Configurational stability studies.** Racemization experiments of axially chiral *N*-vinylquinolinones *Z*−**3a** and *E*−**3a**.

chromatography over silica gel (petroleum ether: ethyl acetate = 50:1 to 10:1) to afford the desired product (*P*,*E*)-**3**. The *Z*/*E* ratio was determined by crude $^1$H-NMR of the reaction mixture.

## Data availability

Crystallographic data for the structures reported in this article have been deposited at the Cambridge Crystallographic Data Centre (CCDC), under deposition numbers CCDC 2082917 [(*P*,*Z*)-**3a**], 2096051 [(*P*,*Z*)-**3m**], 2089101 (**Int-3a**), 2252177 [(*P*,*E*)-**3a**] and 2286999 [(*P*,*Z*)-**5**]. These data can be obtained free of charge from The Cambridge Crystallographic Data Centre via www.ccdc.cam.ac.uk/data_request/ cif. Experimental procedures, characterization of new compounds, and all other data supporting the findings are available in the Supplementary Information. All data were available from the corresponding author upon request. Source data are provided with this paper.

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

## Acknowledgements

We gratefully acknowledge the financial support from the National Natural Science Foundation of China (22201131 for Y.H.) and the Natural Science Foundation of Jiangsu Province (BK20220137 for Y.H.).

## Author contributions

J.W. and J.G. contributed equally to this work. Y.H. designed and directed the project. J.W., J.-Y.Z., M.-J.Z., R.S., Z.Y., and P.-X.X. performed the experiments. J.G. performed the DFT calculations. Y.H. wrote the manuscript with the revisions of all authors.

## Competing interests

The authors declare no competing interests.
