## [Peer Review File · Nature Communications]

Photocatalytic Z/E Isomerization Unlocking the Stereodivergent Construction of Axially Chiral Alkene FrameworksEditorial Note: This manuscript has been previously reviewed at another journal that is not operating a transparent peer review scheme. This document only contains reviewer comments and rebuttal letters for versions considered at *Nature Communications*.

REVIEWER COMMENTS

Reviewer #1 (Remarks to the Author):

This article reports the asymmetric allylic substitution isomerization of synthesized axially chiral N-vinylquinolinones and achieves Z/E isomerization of tetrasubstituted olefins through photocatalysis. The corresponding modifications do not fully address my concerns.

1. In the potential energy surface of Supplementary Fig. 5., it seems to be concluded that the E-configurational olefin is also important product, but this is inconsistent with the experimental Z-configurational olefin as the main product. The energy barrier of conformational isomers 3a1 and 3a2 for interconversion is low, following the Boltzmann distribution. The energy of 3a1 has 1.5kcal lower than 3a2, meaning that 3a1 should be more than ten times to 3a2. Then, after MeONa is added. The process of producing 3a1-1 and 3a2-1 is an exothermic process. At this time, because the conformer interconversion transition state 3aTS-NaOMe has a higher energy than the deprotonation transition state 3a1-TS1 (2.9 kcal/mol vs. 2.4 kcal/mol), furthermore the intermediate 3a1-1 will irreversibly generate a large amount of E-configuration olefin intermediate 3a1-2 (reverse reaction energy barrier is 20.6 kcal/mol) via 3a1-TS1, thus leading to the formation of a large number of E-configuration olefin, which is inconsistent with the experiment. The authors need to carefully evaluate the energy of the corresponding transition states to obtain the self-consistent conclusions for both experiments and calculations.

2. The author did not address my concerns of question 2, regarding the axial chirality effect. I do not find anything useful for that in SI. In case of the importance, I still hope to see some results on that.

3. In the photoexcitation process of Supplementary Fig. 6., the energy difference between S0 state and T1 state is 2.36eV and 2.42eV respectively, which is a large energy difference, and the SOC value of the system is small, making it difficult to realize the ISC process. It is generally believed that the energy difference between the singlet and triplet states is small, and there should be S1 or other singlet states between S0 and T1, so that it has a large spin-orbit coupling and a small energy difference to achieve the ISC process. See Nature 492, 234-238 (2012) and references by the authors. Unfortunately, the changes do not reflect this content. If the author may not an expert for the theoretical chemistry, the relevant discussion could be removed from the paper.

Reviewer #2 (Remarks to the Author):

The study by He and coworkers describes the asymmetric allylic substitution-isomerization (AASI) reaction to access tetra-substituted Z-alkenes with excellent axial chirality. The authors then demonstrate that via selective energy transfer, they can form the alternate E-isomer in appreciable selectivity while retaining axial chirality, despite proceeding through a triplet biradical intermediate. The scope is extensive and the work is well carried out and characterized comprehensively. The mechanistic analysis also provides a firm understanding of the origin of stereoselectivity and the additional work from the previous

version of the study has vastly improved the accessibility of the manuscript. I believe the authors have done an excellent job and believe the study would be well received in Nature Communications. I support publication after some very minor points are addressed below. The authors frequently use scientifically inaccurate language. While this does not detract away from the science it is important that work is communicated with scientific rigor. For example the authors refer to “in both Z- and E- style” I would not refer to these as styles. “in both their Z- and E-configurations” is perhaps more accurate in this instance. In a previous revision of the manuscript I have mentioned that the use of stereochemical editing is not appropriate to describe the enclosed reactivity. Again, I would urge the authors to rephrase this.

Pg 1/Figure 1. The authors describe the transition from Int-I to Int-II as “group switching”. This is very ambiguous and not really descriptive of what occurs. I would suggest bond rotation is more accurate and the authors could draw in the 90 degree rotation as is the case with V later in figure 5.

Pg 2 the authors state “the generated benzylic radical intermediate that may demises the enantioselectivities of the corresponding E products”. I understand what the authors are referring to but perhaps this can be elaborated more clearly to aid the reader. The initial excitation, via energy transfer creates a triplet biradical that ultimately can permit free rotation of the previously unsaturated bond. The key control factor is that this may also permit the rotation of the alpha C-N bond ultimately leading to loss of axially chiral information.

Pg 4 the authors refer to “double bond walking”. It is not clear what the authors are referring to here and this sentence should be rephrased to be accurate.

Figure 5. Like all isomerizations enabled by energy transfer catalysis, the product distribution reaches a photostationary state equilibrium. i.e. one isomer is more efficiently excited than the other and one accumulates over time. Given that Ir(ppy)₃ is higher in energy than both transitions described in figure 5 I believe that the authors should highlight this factor. It is also important to reiterate that IV and V can quite easily return to (P,Z)-3a. As the mechanism is communicated at the moment it would appear all processes are unidirectional, which simply isn't the case.

Figure 5. After an initial excitation. The authors propose a radical distribution to the aromatic ring. Can the authors provide accurate spin density calculations to provide substantial evidence for this? Otherwise it could be conceived that excitation, bond rotation and recombination is so rapid that it supersedes loss of stereochemical information.

Reviewer #3 (Remarks to the Author):

As this is my second or third round of reviewing this paper, I will make my remarks brief. I am still concerned that the low stereochemical stabilities is a weakness of this work, however they have now addressed this head on, and have shown ways to get the barriers to rotation to be 'class 3'. As I mentioned previously, I do like the scaffolds of this work and think they are innovative and have the potential to be pharmaceutically relevant.

I feel this work is OK for this journal, however taking into account the other reviewers comments, which I value, , I wonder if a society journal focusing on organic chemistry could also be a logical place for this paper.

For Reviewer #1:

This article reports the asymmetric allylic substitution isomerization of synthesized axially chiral N-vinylquinolinones and achieves Z/E isomerization of tetrasubstituted olefins through photocatalysis. The corresponding modifications do not fully address my concerns.

1. In the potential energy surface of Supplementary Fig. 5., it seems to be concluded that the E-configurational olefin is also important product, but this is inconsistent with the experimental Z-configurational olefin as the main product. The energy barrier of conformational isomers 3a1 and 3a2 for interconversion is low, following the Boltzmann distribution. The energy of 3a1 has 1.5kcal lower than 3a2, meaning that 3a1 should be more than ten times to 3a2. Then, after MeONa is added. The process of producing 3a1-1 and 3a2-1 is an exothermic process. At this time, because the conformer interconversion transition state 3aTS-NaOMe has a higher energy than the deprotonation transition state 3a1-TS1 (2.9 kcal/mol vs. 2.4 kcal/mol), furthermore the intermediate 3a1-1 will irreversibly generate a large amount of E-configuration olefin intermediate 3a1-2 (reverse reaction energy barrier is 20.6 kcal/mol) via 3a1-TS1, thus leading to the formation of a large number of E-configuration olefin, which is inconsistent with the experiment. The authors need to carefully evaluate the energy of the corresponding transition states to obtain the self-consistent conclusions for both experiments and calculations.

✓ We thank the reviewer for the very positive comments. We have re-optimized the structures of intermediates and transition states. The implicit solvent (toluene) was added in the geometry optimizations to obtain more accurate energy. The energy profiles have been revised in Supplementary Fig. 5a. We now believed that the computational results are consistent with our experimental results.

2. The author did not address my concerns of question 2, regarding the axial chirality effect. I do not find anything useful for that in SI. In case of the importance, I still hope to see some results on that.

✓ We thank the reviewer for the very positive comments. The reason for why the Z-products were obtained rather than E-products was attributed to the relative higher energy profiles of corresponding TS. The energy profiles are now added in the Supplementary Fig. 5b.

3. In the photoexcitation process of Supplementary Fig. 6., the energy difference between S0 state and T1 state is 2.36eV and 2.42eV respectively, which is a large energy difference, and the SOC value of the system is small, making it difficult to realize the ISC process. It is generally believed that the energy difference between the singlet and triplet states is small, and there should be S1 or other singlet states between S0 and T1, so that it has a large spin-orbit coupling and a small energy difference to achieve the ISC process. See Nature 492, 234-238 (2012) and references by the authors. Unfortunately, the changes do not reflect this content. If the author may not an expert for the theoretical chemistry, the relevant discussion could be removed from the paper.

✓ We thank the reviewer for the very positive comments. The discussion of SOC values of

the system on Supplementary Fig. 6 has been removed.

For Reviewer #2:

The study by He and coworkers describes the asymmetric allylic substitution-isomerization (AASI) reaction to access tetra-substituted Z-alkenes with excellent axial chirality. The authors then demonstrate that via selective energy transfer, they can form the alternate E-isomer in appreciable selectivity while retaining axial chirality, despite proceeding through a triplet biradical intermediate. The scope is extensive and the work is well carried out and characterized comprehensively. The mechanistic analysis also provides a firm understanding of the origin of stereoselectivity and the additional work from the previous version of the study has vastly improved the accessibility of the manuscript. I believe the authors have done an excellent job and believe the study would be well received in Nature Communications. I support publication after some very minor points are addressed below.

✓ We thank the reviewer for the very positive comments.

1. The authors frequently use scientifically inaccurate language. While this does not detract away from the science it is important that work is communicated with scientific rigor. For example the authors refer to “in both Z- and E- style” I would not refer to these as styles. “in both their Z- and E-configurations” is perhaps more accurate in this instance.

✓ We thank the reviewer for the comments. The phrase “in both Z- and E- style” has been changed to “in both their Z- and E-configurations”.

2. In a previous revision of the manuscript I have mentioned that the use of stereochemical editing is not appropriate to describe the enclosed reactivity. Again, I would urge the authors to rephrase this.

✓ We thank the reviewer for the comments. The phrase “stereochemical editing” has been changed accordingly.

3. Pg 1/Figure 1. The authors describe the transition from Int-I to Int-II as “group switching”. This is very ambiguous and not really descriptive of what occurs. I would suggest bond rotation is more accurate and the authors could draw in the 90 degree rotation as is the case with V later in figure 5.

✓ We thank the reviewer for the comments. The phrase “group switching” has been changed to “bond rotation”. The int-II of Fig. 1 has been redrawn accordingly.

4. Pg 2 the authors state “the generated benzylic radical intermediate that may demises the enantioselectivities of the corresponding E products”. I understand what the authors are referring to but perhaps this can be elaborated more clearly to aid the reader. The initial excitation, via energy transfer creates a triplet biradical that ultimately can permit free rotation of the previously unsaturated bond. The key control factor is that this may also permit the rotation of the alpha C-N bond ultimately leading to loss of axially chiral information.

✓ We thank the reviewer for the comments. The sentence “the generated benzylic radical intermediate that may demises the enantioselectivities of the corresponding E products”. has been changed to “the initial excitation via energy transfer creates a triplet biradical that may permit free rotation around the chiral axis, thus leading to loss of axially chiral information.”

5. Pg 4 the authors refer to “double bond walking”. It is not clear what the authors are referring to here and this sentence should be rephrased to be accurate.

✓ We thank the reviewer for the comments. The phrase “along with double bond walking” has been changed to “around the C-N axis”.

6. Figure 5. Like all isomerizations enabled by energy transfer catalysis, the product distribution reaches a photostationary state equilibrium. i.e. one isomer is more efficiently excited than the other and one accumulates over time. Given that Ir(ppy)₃ is higher in energy than both transitions described in figure 5 I believe that the authors should highlight this factor. It is also important to reiterate that IV and V can quite easily return to (P,Z)-3a. As the mechanism is communicated at the moment it would appear all processes are unidirectional, which simply isn't the case.

✓ We thank the reviewer for the comments. The energy of Ir(ppy)₃ has been added in the revised manuscript. In Fig. 5, the invertible arrow has been shown from the intermediate II to III.

7. Figure 5. After an initial excitation. The authors propose a radical distribution to the aromatic ring. Can the authors provide accurate spin density calculations to provide substantial evidence for this? Otherwise it could be conceived that excitation, bond rotation and recombination is so rapid that it supersedes loss of stereochemical information.

✓ We thank the reviewer for the comments. The spin density plot of intermediate I is now added in the Supplementary Fig. 6.

Reviewer #3 (Remarks to the Author):

As this is my second or third round of reviewing this paper, I will make my remarks brief. I am still concerned that the low stereochemical stabilities is a weakness of this work, however they have now addressed this head on, and have shown ways to get the barriers to rotation to be 'class 3'. As I mentioned previously, I do like the scaffolds of this work and think they are innovative and have the potential to be pharmaceutically relevant.

I feel this work is OK for this journal, however taking into account the other reviewers comments, which I value, I wonder if a society journal focusing on organic chemistry could also be a logical place for this paper.

✓ We thank the reviewer for the very positive comments.

REVIEWERS' COMMENTS

Reviewer #1 (Remarks to the Author):

Authors had addressed all my concerns. I think the paper now could be published.